# Environmental Robustness and Resilience of Direct-Write Ultrasonic Transducers Made from P(VDF-TrFE) Piezoelectric Coating

**DOI:** 10.3390/s23104696

**Published:** 2023-05-12

**Authors:** Jin Kyu Han, Voon-Kean Wong, David Boon Kiang Lim, Percis Teena Christopher Subhodayam, Ping Luo, Kui Yao

**Affiliations:** Institute of Materials Research and Engineering (IMRE), Agency for Science, Technology and Research (A*STAR), 2 Fusionopolis Way, Innovis #08-03, Singapore 138634, Singapore; hanjk@imre.a-star.edu.sg (J.K.H.); wongvk@imre.a-star.edu.sg (V.-K.W.); limbk@imre.a-star.edu.sg (D.B.K.L.); christopherspteena@imre.a-star.edu.sg (P.T.C.S.); luop@imre.a-star.edu.sg (P.L.)

**Keywords:** structural health monitoring, direct-write, environmental test, piezoelectric ultrasonic transducer, resilience, harsh environment, robustness, piezoelectric polymer PVDF, aircraft ultrasonic

## Abstract

Conformability, lightweight, consistency and low cost due to batch fabrication in situ on host structures are the attractive advantages of ultrasonic transducers made of piezoelectric polymer coatings for structural health monitoring (SHM). However, knowledge about the environmental impacts of piezoelectric polymer ultrasonic transducers is lacking, limiting their widespread use for SHM in industries. The purpose of this work is to evaluate whether direct-write transducers (DWTs) fabricated from piezoelectric polymer coatings can withstand various natural environmental impacts. The ultrasonic signals of the DWTs and properties of the piezoelectric polymer coatings fabricated in situ on the test coupons were evaluated during and after exposure to various environmental conditions, including high and low temperatures, icing, rain, humidity, and the salt fog test. Our experimental results and analyses showed that it is promising for the DWTs made of piezoelectric P(VDF-TrFE) polymer coating with an appropriate protective layer to pass various operational conditions according to US standards.

## 1. Introduction

Current ultrasonic non-destructive testing (NDT) technologies used in areas such as civil structural control [1] and aircraft ultrasonic inspection [2] typically involve manual operation with handheld instrument and are labor-intensive, expensive, and prone to human error. On the other hand, ultrasonic structural health monitoring (SHM) technology, which employs ultrasonic transducers installed in structures, enables continuous inspection without the interruption of service [3]. SHM can provide continuous data for evaluating the evolution of defects or damages as highly demanded in future digital twins and the internet of things (IoT) [4]. However, it is challenging to achieve the desired economic cost and high reliability through the installation of a large number of discrete ultrasonic transducers on the structure to be monitored. Additionally, the bulkiness and weight of the discrete ultrasonic transducers are generally a concern as a significant intrusive factor for structures owners [5]. Therefore, there is a need for a technology that is highly conformable, low profile, lightweight, and can be easily integrated into any surface of the structure to be monitored. It is desirable that this technology can offer non-destructive monitoring functions in real-life operational environments without any interference to the performance of the host structure, as well as being adequately robust and resilient to sustain various harsh environmental conditions.

In previous studies, we reported the application of direct-write piezoelectric ultrasonic transducers (DWTs) for fatigue crack monitoring of aluminum alloy plates [6], defect detection in aluminum plates [7,8], plastic strain monitoring of aluminum alloy plates [9], SHM of steel pipes [10,11], and defect detection in carbon fiber-reinforced polymer plates [12]. These DWTs are made of poly(vinylidenefluoride-co-trifluoroethylene) [P(VDF-TrFE)] that can be fabricated in situ on the desired structure. Through a scalable batch printing process without using any additional adhesive, the cost and fabrication time can be significantly reduced, and it can be conveniently produced on curved surfaces with complex shapes. However, the real-world environmental impacts on the lifetime and performance of these piezoelectric polymer transducers are still uncertain. Industrial applications demand the robustness and resilience of SHM through the continuous exposure to a variety of environments and operating conditions. The continued exposure to service conditions could cause the transducer elements to undergo irreversible changes. In literature, environmental test results of ultrasonic transducers have mainly focused on ceramic piezoelectric elements that require an adhesion layer for applications such as biomedical [13], microelectronics packaging [14], and 3D printing technology [15]. Gao et al. reported the long-term stability in Pb(Zr, Ti)O_3_ (PZT)-based Lamb waves SHM technology [16]. Attarian et al. investigated the reliability of long-term monitoring in a SHM system using PZT [17]. The effect of adhesives and protective coatings on the reliability of the sensor in the PZT system has been reported. There are few studies that evaluated the reliability of piezoelectric polymer sensors under UV radiation and for underwater applications [18,19,20]. However, reports on the robustness and resilience of piezoelectric polymer ultrasonic transducers are lacking, despite the great interest in their applications. The ultrasonic transducers may be exposed to various environments during operations such as temperature changes, humidity, icing, rain, and chemical exposure such as salt fog. These conditions may not significantly affect the structure to be monitored, but they may affect the ultrasonic transducers, resulting in permanent degradation [21] or a temporary false alarm [22]. Thus, it is essential to examine the reversible and irreversible changes to the transducers under various environmental impacts. Knowledge is required to achieve adequately stable transducers that can survive in operating conditions, and to appropriately interpret the ultrasonic signals during the data analysis phase for minimizing incorrect conclusions or false alarms [23].

In this paper, the robustness and resilience of the DWTs made of P(VDF-TrFE) are investigated systematically by exposing the DWTs to various environmental conditions, including temperature changes, humidity, salt fog, icing, and rain. The environmental tests were carried out according to the US standard (MIL-STD-810H) [24].

## 2. Materials and Methods

### 2.1. Direct-Write Ultrasonic Transducer Design

A 1.6 mm-thick aluminum alloy plate with a pair of DWTs operating in the pitch-catch mode was selected as the test coupon for this study. A comb-shaped electrode design was implemented for the DWTs to have excellent Lamb wave mode selectivity. The Lamb wave consists of an anti-symmetric A-mode and a symmetric S-mode [25]. The phase velocity and group velocity dispersion curves of a 1.6 mm-thick aluminum alloy plate were calculated as shown in Figure 1a and 1b, respectively. The electrode design adopts the S1 mode at a frequency of 2.8 MHz as indicated in the figures, and the width of the comb-shaped electrode was determined to be 1.2 mm to match the periodicity of the S1 mode wavelength. 

The schematics of the test coupons are shown in Figure 2a,b. The test coupon was made of an aluminum alloy plate and partially coated with nylon, which is a common material for metal casing protection [26]. Two regions were left uncoated for the P(VDF-TrFE) and the comb-shaped top electrode layers to be coated. The P(VDF-TrFE) layer was coated using a spray coating system and annealed at 135 °C for 30 min. Thereafter, the P(VDF-TrFE) coatings were poled using a corona discharge gun. The comb-shaped top electrode was then spray coated on the P(VDF-TrFE) coating with the help of a patterned sticker mask, as shown in Figure 2c. A pair of customized flexible printed circuit (FPC) that come together with an SMA connector (SubMiniature version A) were installed on the comb-shaped top electrode using silver-filled conductive epoxy. Sealant (Nippon Vinilex 5170, Tokyo, Japan) was applied to the DWTs as a protective layer as shown in Figure 2d. Next, SMA cables were connected to the coupon and secured by wrapping parafilm around the connection. Lastly, a conformal coating (ELEPCOAT LSS-520MH, Nitto, Osaka, Japan) was applied to the SMA connection to protect it against environmental conditions. 

The pitch-catch mode was conducted by connecting the ultrasonic transmitter and receiver to an ultrasonic testing system (Ultratek ultrasonic system). The ultrasonic testing system sent an excitation signal (3-cycle sine wave with 100 Vpp with a frequency of 2.8 MHz) to the ultrasonic transmitter for the generation of the Lamb wave that travels along the test coupon and is detected by the ultrasonic receiver, which is positioned 20 mm away from the ultrasonic transmitter. The gated ultrasonic signal shown in Figure 2e is the S1 mode generated and detected by the DWTs in pitch-catch mode. During the environmental tests, the ultrasonic testing system obtained one set of ultrasonic signals every 5 min. Each of the collected ultrasonic signals was the product of 512 waveform averages. The absolute peak amplitude of the gated ultrasonic signal was then computed for further analysis. 

The piezoelectric properties of the fabricated P(VDF-TrFE) were measured using a laser scanning vibrometer (LSV), as shown in Figure 2f [27]. The inset of Figure 2f presents the piezoelectric response of the DWTs and the displacement that occurred according to the pattern of the electrode. Determined from the line displacement-profile, each displacement is about 150 pm, and their effective d_33_ value is about −15 pm/V, which is consistent with the previous work on direct-write P(VDF-TrFE) transducers [7,9].

### 2.2. Environmental Test 

Figure 3 presents a graph showing the settings used in various environmental conditions including temperature, humidity, salt fog, icing, and rain that refers to US Standard (MIL-STD-810H). A stability test was carried out to establish a baseline in measurement using three test coupons (labelled as #1, #2, and #3). The stability test was conducted at 25 ± 0.2 °C for three days, as shown in Figure 3a, in a temperature chamber (ESPEC SU-240, Hudsonville, MI, USA, shown in the inset). This was then followed by a combined temperature test to examine the effects of high and low temperatures, as shown in Figure 3b. Next, the effect of a humid atmosphere was examined, as shown in Figure 3c. This test was carried out in an environmental test chamber, as shown in Figure 3c inset (Memmert CTC256, Schwabach, Germany). For this test, the humidity was controlled in the range of 30 to 95% using de-ionized water, and all the water that condensed in the chamber was drained without reuse. After that, the salt fog test was carried out next to evaluate the effectiveness of protective coatings and finishes. Test coupons were placed in the salt spray test chamber shown in Figure 3d inset (Model YSST-108, Guangdong Yuanyao Test Equipment Co., Ltd., Guangdong, China), at a tilted angle of 45 degrees to avoid pooling of water from condensation. The salt fog test was carried out following the conditions shown in Figure 3d by using a 5% sodium chloride solution. A fallout rate of 1 to 3 mL/h/80 cm^2^ was used, and the pH range of the fallout solution was determined to be between 6.5 and 7.2. The fallout was retrieved from collection cups located both near and far from the spray tower. 

Another three test coupons (#4, #5, and #6) were used for the icing and rain tests. The icing test was carried out using a temperature chamber (ESPEC SU-240) to evaluate the impact of coupons exposed to icing caused by freezing rain or freezing drizzle. Figure 3e is a temperature–time graph obtained to confirm the effect of forming ice on the coupon. For ice formation on the test coupons, a template using polyethylene terephthalate (PET) was used, as shown in the inset of Figure 3e. The height of the acrylic template is 13 mm, which is for ice thicknesses under normal conditions indicating moderate loads. The type of ice required transparent glazed ice, and for this purpose, water was cooled to 0 °C before being filled into the acrylic template on the test coupons. As a result, transparent ice was successfully obtained, as shown in the inset of Figure 3e. The test coupons #4, #5, and #6 were then used in the rain test. This test was carried out to investigate the effect of the protective layer on rain and its effect on robustness and resilience. For the rain test, a rain simulator was customized and fabricated to simulate exaggerated rain conditions, as shown in the inset of Figure 3f. The rain simulator is made of a 276 kPa nozzle pressure with a flow rate of 20.8 L/min, corresponding to a rainfall velocity of approximately 64 km/h. Figure 3f is a graph showing the flow rate of water over time. 

## 3. Results

Figure 4 is a graph showing the peak voltage values of the gated ultrasonic signals obtained during the temperature stability, combined temperature, humidity, and salt fog tests. The peak voltage values of the gated ultrasonic signals are presented in the form of a box plot, showing the mean value, 25 and 75 percent values, and two whiskers showing the minimum and maximum values. Figure 4a shows a set of ultrasonic peak voltage box plots for the three samples during the temperature stability test. The ultrasonic peak voltage for each sample was stable for 3 days. Another set of ultrasonic peak voltage box plots for each of the test coupons measured under the combined temperature test is given in Figure 4b. From the combined temperature test, the peak voltage changed with the temperature. This is due to the effect of temperature on the Lamb waves that is normally compensated in SHM systems [28]. Nonetheless, comparing ultrasonic peak voltages in the initial *i* step and the final *iv* step, it is observed that the ultrasonic peak voltage remains stable after the combine temperature test. Therefore, it can be concluded that the DWT’s performance in generating and detecting Lamb wave was not substantially affected by the combined temperature conditions. 

In the next test, the ultrasonic peak voltage in Figure 4c shows a very large spread at 75 to 95% humidity. The large spread of ultrasonic peak voltage is due to condensation of water on the sample. After three repetitions of humidity tests, the ultrasonic peak voltage at the final *iii* step is smaller than that at the initial *i* step. This suggested that the ultrasonic peak voltage decreased by about 10 to 20% after the humidity test. The causes for the decrease in ultrasonic peak voltage are explained with impedance and piezoelectric constant analysis in the next paragraph. Figure 4d summarizes the ultrasonic peak voltages obtained from the salt fog test for the three coupons. As soon as test coupons #1 and #3 were exposed to salt fog, the ultrasonic peak voltage decreased to close to 0. This is because to contact of the salt test solution on the coupon. The condensed solution on the test coupon causes changes in the boundary condition that leads to ultrasonic energy leakage to the liquid [11]. Nonetheless, the ultrasonic peak voltage recovered after the drying and cleaning processes. However, in the case of test coupon #2, the ultrasonic peak voltage did not restore to its original value. This is a strong indication that an issue has occurred in coupon #2 and will be further investigated. 

Impedance, dielectric loss, and piezoelectric constant for DWTs on test coupons were measured before and after each environmental test. Figure 5 presents the trend of impedance, dielectric loss, and piezoelectric coefficient for the coupons. In the case of P(VDF-TrFE), the dielectric losses are mainly determined by the DC conduction of free charge at low frequency and molecular motion at high frequency [29]. The electrical impedance is directly related to the mechanical impedance of the host component to which the P(VDF-TrFE) is attached [30]. If there is damage such as delamination, it can be detected through the electrical impedance analysis. Through this impedance analysis, it can be confirmed how the state of the interface has changed after environmental testing.

Figure 5a,b show the dielectric loss and impedance data measured on the receiver and transmitter sides, respectively, before testing. The impedance of P(VDF-TrFE) has a resistance of several MΩ at low frequencies and decreases with increasing frequencies as the space charge polarization contributes less [31]. Figure 5c–e are graphs showing the average values of the electrical characteristics of all coupons after each experiment. Impedance and dielectric loss values were taken at 2.8 MHz using ultrasonic measurement. The piezoelectric coefficient was obtained by averaging the values obtained by line profiling the three electrodes of each sample, as shown in Figure 2f. 

Figure 5c,d show the average impedance values and average dielectric loss values, respectively, after each test condition. In the temperature stability, combined temperature (marked as CT in the graph), and humidity tests, the impedance was constant at about 250 Ω with a small standard deviation. Likewise, the dielectric loss was also constant at about 0.16 with a small standard deviation. However, after the salt fog test, the impedance increased while the dielectric loss decreased, and the standard deviation for both widened. Figure 5e shows the average values of effective d_33_ to be stable between −15 and −18 pm/V after each experiment. 

Figure 5f–h present maps showing the percentage change of the value obtained before (*x*_1_) and after (*x*_2_) the experiment for each coupon. The percentage change Δ*x* was obtained using the following formula:(1)$$\Delta x=\frac{x_{2}-x_{1}}{x_{1}}\times 100$$

A percentage change of around 10% was observed in the values of impedance, dielectric loss, and piezoelectric constant after each test condition, but after the salt fog test, a few coupons showed an unusually high percentage change, particularly in coupon #2. In the case of the piezoelectric coefficient, it was even not measurable for 2T (referring to the d_33_ values on the transmitter side of coupon #2). These observations imply that coupon #2 is damaged. In the impedance-mechanical method [32], electro-mechanical impedance technology for damage detection utilizes changes in structural impedance to determine damage to a structure. When a change occurs in the interlayer, the physical and mechanical properties of the structure will change due to the discontinuity at interfaces. The mechanical properties of the structure can be detected with high sensitivity by measuring the electro-mechanical impedance signatures of a piezoelectric transducer. Therefore, the change in the impedance and dielectric loss of the piezoelectric transducer that occurred after the salt fog test implies a change in the host structure and the interface between the host structure and the piezoelectric transducer.

After the salt fog test, coupon #2 was examined with optical microscopy and field-emission scanning electron microscopy (FESEM) to find out what happened to the sample, as shown in Figure 6. The blisters were visible in coupon #2, as shown in Figure 6a. Figure 6b is an enlarged view of the red circled area of Figure 6a by tilting, and empty space is evident at the edge due to the blisters. This part is near the red arrow in the inset of Figure 6b and corresponds to the edge part of the P(VDF-TrFE) layer. It is obvious that this defect poses a vulnerable area for environmental exposure. In order to observe how this blister region formed, piezoelectric displacement mapping before and after the salt fog test was compared, as shown in Figure 6c–f. Figure 6c,d show optical images and piezoelectric displacement mapping data, respectively, of the coupon obtained after the humidity test (C-Humidity). The significant changes in the morphology and piezoelectric displacement suggest that delamination happened in the regions marked by the arrow. Despite these changes, there were no large changes in the ultrasonic and electrical signals of C-Humidity, as shown in Figure 4 and Figure 5. Figure 6e,f are optical images and piezoelectric displacement mapping data, respectively, of the coupon obtained after the salt fog test (C-Saltfog). In addition to the previous delamination around the electrode, further delamination was observed in another area marked by the red arrow. The ultrasonic signal was greatly impacted by this increased area of delamination.

In order to examine the delamination through the cross-sectional view, coupons before and after the salt fog test were cut using a saw blade. The cutting was made around the red dashed line in Figure 6a. Figure 6g presents a cross-sectional optical image of C-Humidity, showing that P(VDF-TrFE) and sealant were well coated on the aluminum alloy. In the FESEM image in Figure 6i, P(VDF-TrFE) and sealant layers were identified, and the layers adhered well without obvious separation. This shows that these layers were not separated by the mechanical force generated from cutting with the saw blade. Figure 6h is a cross-sectional optical image of C-Saltfog, showing blisters in the P(VDF-TrFE) and sealant layers on aluminum alloy. In the enlarged FESEM image of Figure 6j, it is observed that the layers are separated from each other. The delamination could change the piezoelectric displacement, dielectric properties, and impede the generation and detection of ultrasonic signals. Considering only one of the three coupons suffered from the failure, the delamination formed by the salt fog test might be caused by imperfect coating processing in the current transducer design rather than a material-intrinsic issue. Since this delamination started at the edge of the coupon, we believe that this issue can be sufficiently prevented by the implementation of a suitable protective layer or solved by improving the fabrication process for eliminating the defects at the edge.

Since partial delamination and changes in electrical properties were observed after salt fog exposure for coupon #2, the follow-up icing and rain tests were conducted with a new group of test coupons (#4, #5, and #6). According to the ultrasonic peak voltage obtained during the icing and rain tests, as presented in Figure 7a and Figure 7b, respectively, all the signals went through large changes during the icing and rain processes. These changes in ultrasonic signals are mainly due to the presence of water on the test coupons as well as a variation in temperature [23,28]. Nevertheless, they all recovered to the same magnitude as the initial signal after the experiments. This shows that the DWTs are able to withstand the simulated icing and rain conditions.

After the icing and rain exposure experiments, the average values of impedance, dielectric loss, and piezoelectric constant were measured, as shown in Figure 8a–c. The average impedance was constant at around 260 Ω. Compared to the initial, the average dielectric loss of the icing and rain test slightly decreased from 0.17 to 0.16, and the standard deviation showed an increasing trend. The average effective d_33_ value was almost constant at −18 pm/V. Figure 8d–f shows the percentage change of the impedance, dielectric loss, and piezoelectric coefficient for each coupon, and they had a percentage change of less than 10%, indicating that our DWTs are robust for rain and icing exposure.

## 4. Conclusions

The robustness and resilience of the ultrasonic structural health monitoring using direct-write ultrasonic transducers (DWTs) made from P(VDF-TrFE) piezoelectric coating were studied by observing the gated ultrasonic signal magnitude and the electrical properties after each environmental testing carried out referring to the US standard (MIL-STD 810H). The experimental results indicated that the DWTs were successfully able to pass the testing of high and low temperature cycles, rain, humidity, icing, and salt fog. After the salt fog tests, visual and electrical property mapping showed possible structural failure and property degradation in some of the coupons, but not in others. Close morphological observation of a degraded coupon revealed that delamination was initiated from the edge of the piezoelectric films and protective layers and extended to the region under the electrode coverage to generate the ultrasonic signals. With most of the samples passing all the relevant testing, it is promising for the DWTs made from P(VDF-TrFE) to meet the US standard (MIL-STD 810H), showing the feasibility of their practical applications.

## Figures and Tables

**Figure 1 sensors-23-04696-f001:**
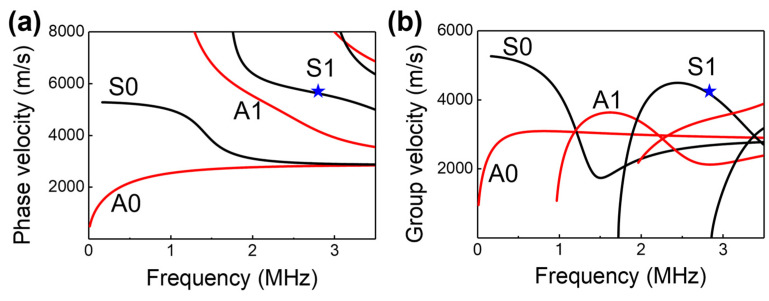
Dispersion curves of an aluminum alloy plate with a thickness of 1.6 mm, phase velocity (**a**) and group velocity (**b**) versus frequency.

**Figure 2 sensors-23-04696-f002:**
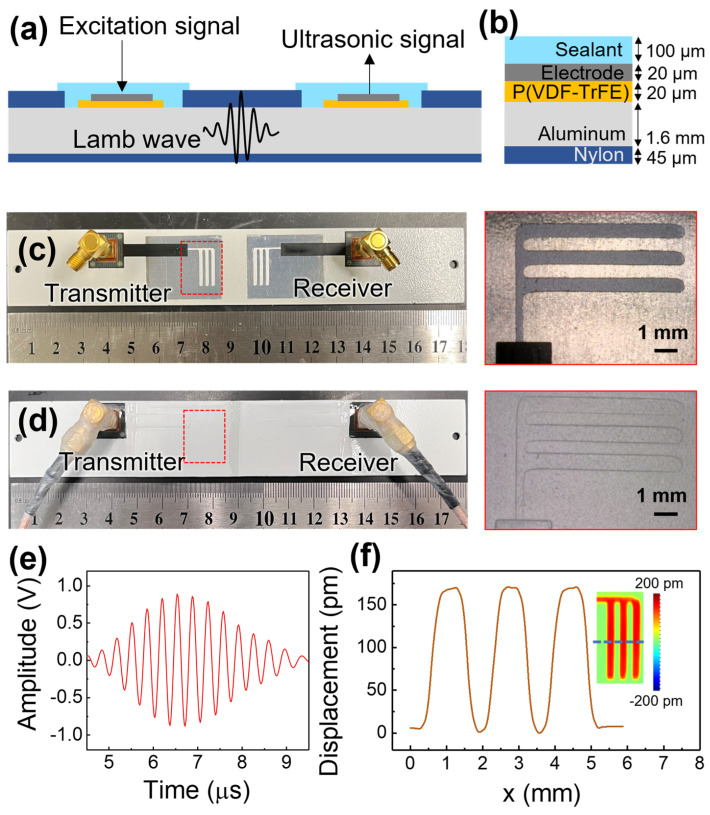
Schematic showing (**a**) coupon structure and (**b**) thickness of the layers, (**c**) photo of the coupon after coating P(VDF-TrFE) and electrodes, and (**d**) photo after coating a protective layer, (**e**) received ultrasonic signal measured via the pitch-catch mode, and (**f**) displacement versus location measured at 20 Vpp, 100 kHz; inset: effective d_33_ mapping, measured with LSV.

**Figure 3 sensors-23-04696-f003:**
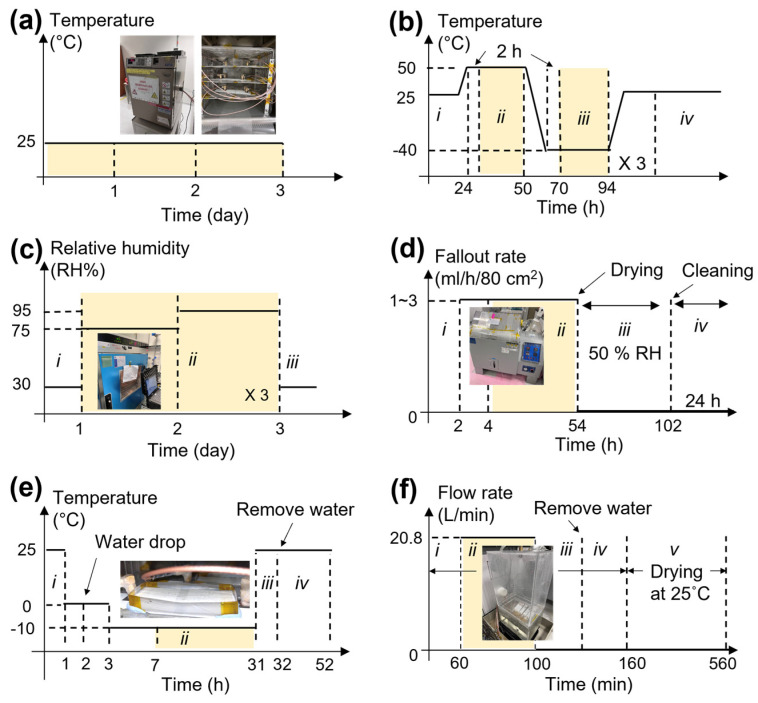
Graphs showing testing conditions in various environments referring to the US Standard MIL-STD 810H, for the (**a**) temperature stability, (**b**) combined temperature: (*i*) an initial step at room temperature, (*ii*) 50 °C, (*iii*) −40 °C, and (*iv*) final step, (**c**) humidity: (*i*) an initial step at 30RH%, (*ii*) 75, and 95 RH%, and (*iii*) final step at 30 RH%, (**d**) salt fog: (*i*) an initial step, (*ii*) during salt fog, (*iii*) drying step after stop salt fog, and (*iv*) drying step after cleaning salt crystals on plate, (**e**) icing: (*i*) an initial step, (*ii*) formation step of icing, (*iii*) increasing temperature, and (*iv*) after removal of the water on the plate, and (**f**) rain tests: (*i*) initial step, (*ii*) during water flow, (*iii*) no water flow, (*iv*) removal of water, and (*v*) drying. Colored regions indicate the duration of each environmental test.

**Figure 4 sensors-23-04696-f004:**
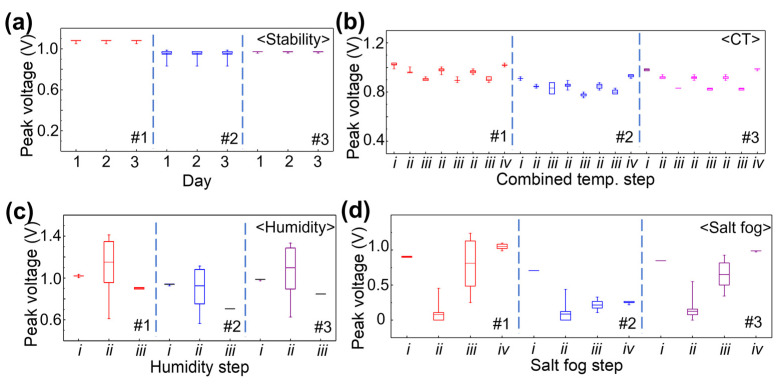
Sets of boxplot for each of test coupons (#1, #2, and #3) showing the ultrasonic peak voltage obtained at each stage for the: (**a**) temperature stability test, (**b**) combined temperature (CT) test (*i*: initial stage at RT, *ii*: high temperature, *iii*: low temperature, *iv*: final stage at RT), (**c**) humidity test (*i*: initial stage at 30 RH%, *ii*: humidity at 75~95 RH%, *iii*: final stage at 30 RH%), and (**d**) salt fog test (*i*: initial stage, *ii*: during salt fog, *iii*: dried at chamber, *iv*: cleaned and dried at 25 °C), box plot: box showing the mean value, 25 and 75 percent values of the total data, and two whiskers showing the minimum and maximum values.

**Figure 5 sensors-23-04696-f005:**
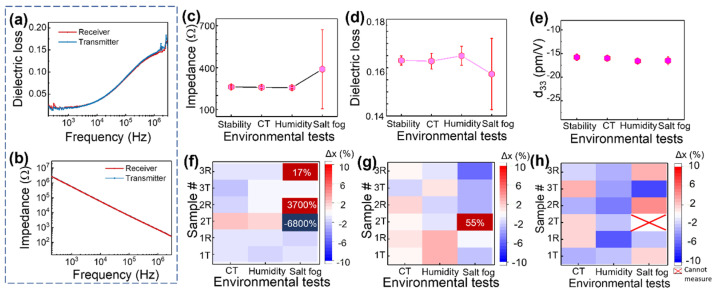
(**a**) Dielectric loss and (**b**) impedance as a function of frequency measured before the stability test for P(VDF-TrFE) on receiver and transmitter sides of coupon #1; graphs showing average values of (**c**) impedance, (**d**) dielectric loss, and (**e**) effective d_33_ measured after each experimental test, and mapping showing percentage change in (**f**) impedance, (**g**) dielectric loss, and (**h**) effective d_33_ values obtained before and after the experiment for test coupons #1, #2, and #3. (CT: Combined temperature) Impedance and dielectric loss were measured at 2.8 MHz, and effective d_33_ was measured at 100 kHz.

**Figure 6 sensors-23-04696-f006:**
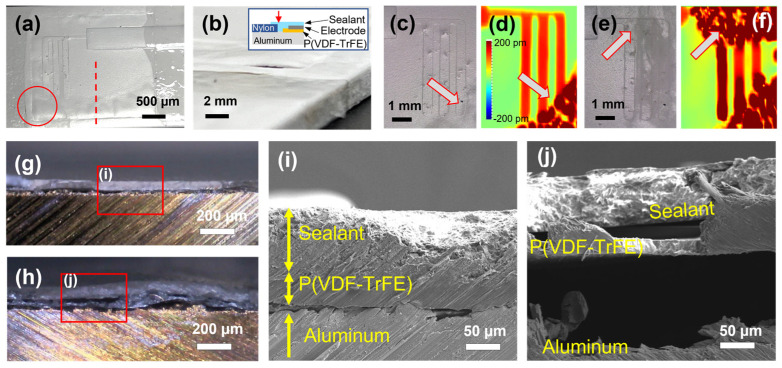
(**a**,**b**) Photos of coupon #2 and (**c**) optical microscope image and (**d**) piezoelectric displacement mapping of coupon #2 after the humidity test; (**e**) optical microscope image and (**f**) piezoelectric displacement mapping of coupon #2 after the salt fog test; optical microscope image of coupon #2 after the (**g**) humidity test, (**h**) salt fog test; FESEM image of coupon #2 after the (**i**) humidity test, (**j**) salt fog test.

**Figure 7 sensors-23-04696-f007:**
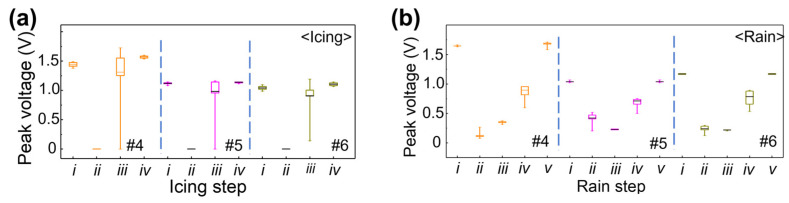
Boxplot graphs showing the mean, maximum, minimum, 25, and 75% values of the peak voltage of the ultrasonic signals at each stage of (**a**) icing (*i*: initial stage, *ii*: during icing, *iii*: melted ice, and *iv*: dried) and (**b**) rain test (*i*: initial stage, *ii*: during raining, *iii*: stop raining, *iv*: remove water, and *v*: dried), for test coupons #4, #5, and #6.

**Figure 8 sensors-23-04696-f008:**
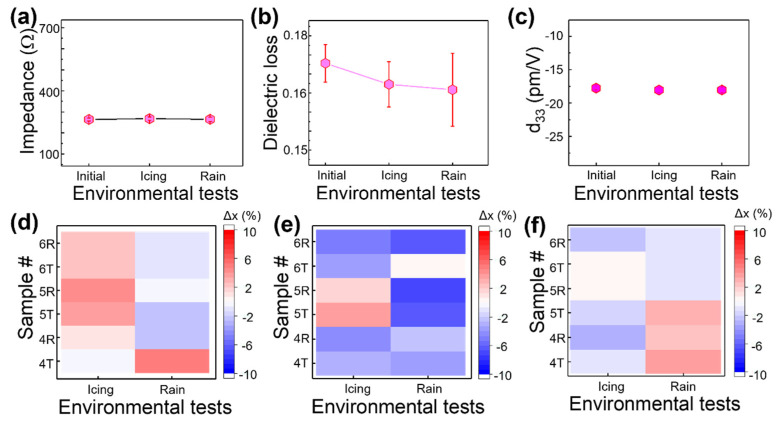
A graph showing the average value of (**a**) impedance, (**b**) electrical loss, and (**c**) the effective d_33_ value measured after each icing and raining experimental test; a mapping showing the percentage of the change obtained before and after the experiment in (**d**) impedance, (**e**) electrical loss, and (**f**) effective d_33_ values for each of the test coupons #4, #5, and #6. Impedance and dielectric loss were measured at 2.8 MHz, and effective d_33_ was measured at 100 kHz.

## Data Availability

Not applicable.

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
