# Peer review of "Environmental Robustness and Resilience of Direct-Write Ultrasonic Transducers Made from P(VDF-TrFE) Piezoelectric Coating"

_sensors, 2023, doi:10.3390/s23104696_

Round 1

Reviewer 1 Report

This paper present the experimental results of transducer under various environment. The experimental results are quite interesting. However, there are minor points for the revision.

- There are graphical error in Figure 3. It should be modified. 

- The experimental results should more extensively explained with scientific support.

The English is over all fine in the manuscript. 

Reviewer 2 Report

This is a relatively well-written manuscript. Authors may consider adding experiments to detect a flaw in the aluminum as a function of PVDF aging. Afterall, that is the most important task of ultrasonic method. 

Authors may consider having the manuscript edited by a native English speaker.

Reviewer 3 Report

This paper investigates environmental robustness and resilience of direct-write ultrasonic transducers (DWTs) made from P(VDF-TrFE) piezoelectric coating. The ultrasonic signals of the DWTs and properties of the piezoelectric polymer coatings fabricated in situ on the test coupons were evaluated during and after exposure to various environmental conditions, including high and low temperatures, icing, rain, humidity, and salt fog test.

This study provides novel and interesting results. In addition, it is very well organized. Meanwhile, the authors should consider the following issues:

-There is a marked revised paper which is attached below. The authors should consider the required revisions and comments. There are some grammar. and typing errors that should be corrected. The addition of the following benchmark study can help to enrich the References part: 

- "A European Association for the Control of Structures joint perspective. Recent studies in civil structural control across Europe". Structural Control and Health Monitoring. 2014; 21(12):1414-36. https://doi.org/10.1002/stc.1652

Overall, this paper can be accepted for publication after the completion of required revisions and comments. 

There is a marked revised paper which is attached below. There are some grammar. and typing errors that should be corrected. 

Reviewer 4 Report

1. A review of the literature is, in my opinion, insufficient. The authors cited only 4 items (8-12) in which similar topics were dealt with.

2.Conclusion would be much clearer if it was in points. Also, there is information that I feel should have been in the previous chapter, not in concusion chapter.
